# Automatic Differentiation of Programs with Discrete Randomness

**Gaurav Arya**
Massachusetts Institute of Technology, USA
`aryag@mit.edu`

**Moritz Schauer**
Chalmers University of Technology, Sweden
University of Gothenburg, Sweden
`smoritz@chalmers.se`

**Frank Schäfer**
Massachusetts Institute of Technology, USA
University of Basel, Switzerland
`franksch@mit.edu`

**Chris Rackauckas**
Massachusetts Institute of Technology, USA
Julia Computing Inc., USA
Pumas-AI Inc., USA
`crackauc@mit.edu`

## Abstract

Automatic differentiation (AD), a technique for constructing new programs which compute the derivative of an original program, has become ubiquitous throughout scientific computing and deep learning due to the improved performance afforded by gradient-based optimization. However, AD systems have been restricted to the subset of programs that have a continuous dependence on parameters. Programs that have discrete stochastic behaviors governed by distribution parameters, such as flipping a coin with probability $p$ of being heads, pose a challenge to these systems because the connection between the result (heads vs tails) and the parameters ($p$) is fundamentally discrete. In this paper we develop a new reparameterization-based methodology that allows for generating programs whose expectation is the derivative of the expectation of the original program. We showcase how this method gives an unbiased and low-variance estimator which is as automated as traditional AD mechanisms. We demonstrate unbiased forward-mode AD of discrete-time Markov chains, agent-based models such as Conway's Game of Life, and unbiased reverse-mode AD of a particle filter. Our code package is available at `https://github.com/gaurav-arya/StochasticAD.jl`.

## 1 Introduction

Automatic differentiation (AD) is a technique for taking a mathematical program $X(p)$ and generating a new program $\widetilde{X}(p) = \frac{\mathrm{d}X}{\mathrm{d}p}$ for computing the derivative [1, 2]. AD is widely used throughout machine learning and scientific computing due to the increased performance of gradient-based optimization techniques compared to derivative-free methods [3]. However, if $X(p)$ returns the flip of a coin with probability $p$ of receiving a 1 and probability $1-p$ of receiving a 0, it is clear that $\frac{\mathrm{d}X}{\mathrm{d}p}$ is not defined in the classical sense. But when attempting to calibrate the parameter $p$ to data, one may wish to fit the model using statistical quantities, e.g. find $p$ such that the average of $X(p)$ is close to the average sum of $N$ real-world coin flips. Given this use case, can one automatically construct a program $\widetilde{X}(p)$ that computes the derivative of the statistical quantities, i.e. $\mathbb{E}[\widetilde{X}(p)] = \frac{\mathrm{d}\mathbb{E}[X(p)]}{\mathrm{d}p}$?

A naïve solution to this problem would be to use finite differences, i.e.:

$$\frac{\mathrm{d}\mathbb{E}[X(p)]}{\mathrm{d}p} \approx \frac{\mathbb{E}[X(p+\varepsilon)] - \mathbb{E}[X(p)]}{\varepsilon}. \tag{1.1}$$

36th Conference on Neural Information Processing Systems (NeurIPS 2022).

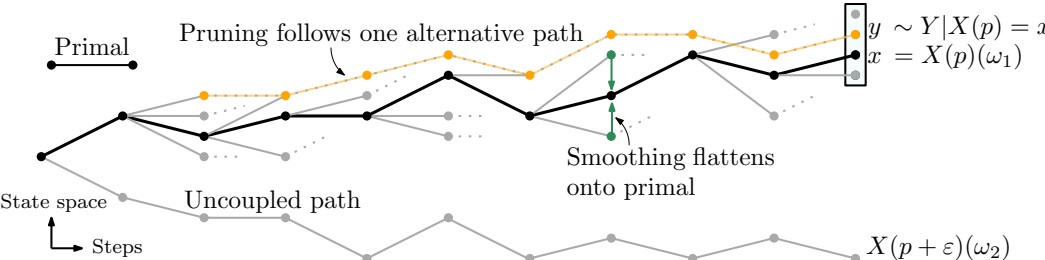

Figure 1: Qualitative sketch of our method and comparison to finite differences. The primal computation (solid black line) samples $X(p)$ with random number sequence $\omega_1$. Black-box finite differences samples the perturbed program $X(p + \varepsilon)$ with an *independent* random number sequence $\omega_2$ (bottom) for some *finite* choice of $\varepsilon$. In contrast, the component $Y$ of the stochastic derivative of the program (Section 2) considers the effect of the minimal possible perturbations (gray lines diverging from primal path) to the original program which could stem from a truly *infinitesimal* change in the input $p$.

Finite differences' major issue in the context of stochastic programs is that this calculation does not correlate the calculation of $\mathbb{E}[X(p+\varepsilon)]$ with the calculation of $\mathbb{E}[X(p)]$. This leads to a large variance in the finite-difference estimator [4] that goes to $\infty$ as $\varepsilon \to 0$. This issue of unbounded variance could be solved if one could run the perturbed program with the same set of random numbers and directly estimate $\mathbb{E}[X(p + \varepsilon) - X(p)]$ by a method that is well-posed in the limit of $\varepsilon \to 0$ (Fig. 1).

We demonstrate how to automatically construct a new stochastic program $\widetilde{X}(p)$ whose expectation satisfies $\mathbb{E}[\widetilde{X}(p)] = \frac{\mathrm{d}\mathbb{E}[X(p)]}{\mathrm{d}p}$. We derive this through a technique which we term the *stochastic derivative*, propagating the proportional probability of differing event outcomes due to infinitesimal changes in $p$. Our technique has the following design goals:

- **Composability.** We differentiate stochastic programs $X(p)$ which are themselves composed of many "elementary" stochastic programs, including samples from discrete distributions such as the Bernoulli, Poisson, and Geometric, and samples from continuous distributions. These may be composed (chained, added, etc.) arbitrarily, with computational cost independent of how they are composed.

- **Unbiasedness.** The program's discrete structure is preserved, leading to a provably unbiased estimator that avoids continuous relaxations and tunable or learned internal parameters.

- **Low variance.** We consider correlated paths through the computation that are linked by the smallest possible perturbation (Fig. 1), generalizing the widely-used pathwise gradient estimator for continuous randomness [5] to the discrete case.

We show the utility of our technique by demonstrating forward-mode AD of stochastic simulations like inhomogeneous random walks and agent-based models such as the Game of Life. To achieve $\mathcal{O}(1)$ computational overhead for these applications, we combine stochastic derivatives with an online strategy we call *pruning*. We also demonstrate a straightforward way to perform reverse-mode AD from our approach via *smoothing*, allowing us to derive from first principles the biased but empirically successful straight-through gradient estimator [6] as a special case and to construct an unbiased end-to-end reverse-differentiable particle filter, recovering a technique discovered in [7]. We provide an open-source implementation of the method, `StochasticAD.jl`, for readers to explore the technique on their own applications.

## 1.1 Related work

Gradient estimators for stochastic functions can be divided into three different classes [5]: score-function [8, 9, 10], measure-valued [11, 12], and pathwise [13, 14] gradient estimators. When it comes to discrete randomness, the score-function method is a popular general-purpose choice because it is unbiased and can be composed through stochastic computation graphs [15]. However, the score-function method does not search for correlated paths through the computation and thus suffers from high variance, making gradient computation with discrete variables challenging. A number

of techniques (e.g. REBAR [16], RELAX [17]) have been introduced that rely on control variates for variance reduction [18]. Measure-valued derivatives also have no notion of intrinsic coupling, though coupling can be achieved using common random numbers for certain distributions [19]. As an alternative direction, Gumbel-Softmax [20] considers a continuous relaxation of discrete programs so that a pathwise gradient estimator, based on the "reparameterization trick" [14], can be applied. However, such methods face a bias-variance tradeoff and are inapplicable to discrete programs that cannot be continuously relaxed. Our stochastic derivatives also extend the pathwise gradient estimator to discrete programs but do so unbiasedly. The conceptual starting point of our approach is finite differences with common random numbers [21, 22, 23], whose ideas have also been extended by direct optimization [24, 25], but crucially we show how to take the exact limit of step size $\varepsilon \to 0$ even in the discrete case. The field of smoothed perturbation analysis [26, 27] develops a mathematically equivalent object to our smoothed stochastic derivative based on conditional expectations, which is a special case of our formalism, and has also considered a randomized approach similar to our pruning technique in the context of generalized semi-Markov processes [28]. However, these ideas have not previously been applied to construct a general-purpose AD method via rigorous composition rules and the algorithms and data structures to realize the approach automatically.

Although our method can be used to hand-derive a gradient estimator, the main feature is composability through user-written functions, enabling an automated mechanism. While mainstream AD frameworks do not support unbiased differentiation of discrete random programs, Storchastic [29] is a specialized framework for AD of stochastic computation graphs [15] where the user can specify which estimator to use at each node, as well as any tunable hyperparameters. Storchastic implements an exhaustive set of prior gradient estimation methods at each sampling step. However, the runtime of the derivative estimate is in general exponential in the length of the largest chain of stochastic nodes, an artifact of the way many prior gradient estimators compose [29, 30]. In Section 3 we demonstrate stochastic AD that matches the computational complexity of deterministic AD even when discrete random functions are chained together, alleviating these performance issues.

## 2 Composable derivatives of stochastic programs

In this section, we develop the notion of a *stochastic derivative* for programs containing discrete randomness. We shall motivate this object as a natural generalization of the pathwise gradient estimator for continuous random programs, and present the key ideas underpinning the formalism. When describing infinitesimal asymptotics, we say a function $g(\varepsilon)$ is $\mathcal{O}(\varepsilon)$ if $|g(\varepsilon)| \leq C|\varepsilon|$ for some real $C$ and all sufficiently small $\varepsilon$. Colloquially, we describe quantities that are $\mathcal{O}(\varepsilon)$ as "infinitesimal".

### 2.1 Infinitesimally perturbing a stochastic program

We are interested in differentiating stochastic programs, formally defined below. A stochastic program can be thought of as a map from an input $p$ to a random variable $X(p)$. Here, $X(p)$ can represent either an "elementary" program such as a draw from a Bernoulli distribution, or the full user-provided stochastic program represented using many elementary programs, e.g. a simulation of a random walk.

**Definition 2.1.** A *stochastic program* $X(p)$ is a stochastic process with values in a Euclidean space $E$, whose index set $I$ is either an open subset of a Euclidean space or a closed real interval.

Let $\Omega$ be the sample space, equipped with a probability distribution $\mathbb{P}$. To sample $X(p)$ at input $p \in I$, which we call the *primal evaluation*, one should imagine a sample $\omega$ being randomly chosen from $\Omega$ according to $\mathbb{P}$ to produce an output $X(p)(\omega) \in E$. Note that $\mathbb{P}$ is independent of $p$ and $X(p)$ is a map $\Omega \to E$; such a formulation has been called the "reparameterization trick" [14]. For example, a Bernoulli distribution $\mathrm{Ber}(p)$ can be represented by choosing a uniform random $\omega \in [0, 1]$ and defining $X(p)(\omega) = \mathbf{1}_{[1-p,1]}(\omega)$, where $\mathbf{1}_S$ is the indicator function for a set $S$ (see Fig. 2b).

At fixed $p \in I$ we define the differential $\mathrm{d}X(\varepsilon)$, which is itself a stochastic program:

$$\mathrm{d}X(\varepsilon) = X(p+\varepsilon) - X(p). \tag{2.1}$$

Let us now restrict our attention to the case where $I$ is a closed real interval, so that $p, \varepsilon \in \mathbb{R}$. The sensitivity of stochastic programs $Z$ with more general index sets can be understood at an input $\mathbf{u}$ by studying at $p = 0$ the directional perturbation $X(p) = Z(\mathbf{u} + p\mathbf{v})$ in a direction $\mathbf{v}$.

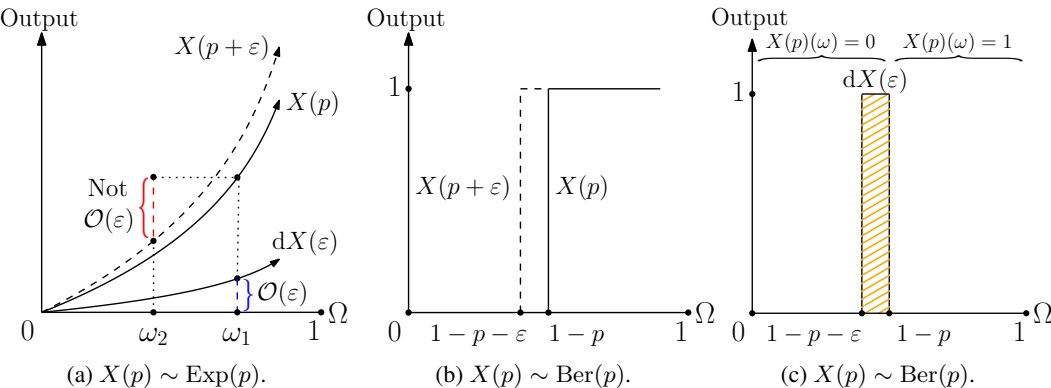

(a) $X(p) \sim \text{Exp}(p)$.        (b) $X(p) \sim \text{Ber}(p)$.        (c) $X(p) \sim \text{Ber}(p)$.

Figure 2: Illustrations of $dX(\varepsilon)$ for a small finite $\varepsilon$, assuming $\varepsilon > 0$ for simplicity. **(a)** The $\mathcal{O}(\varepsilon)$ range of $dX(\varepsilon)$ for continuous $X(p)$. The blue bracket indicates a sample from $dX(\varepsilon)$, while the red bracket indicates a sample obtained by black-box finite differences. **(b)** The original program $X(p)$ and the perturbed program $X(p + \varepsilon)$ for discrete (Bernoulli) $X(p)$. **(c)** Only those samples $\omega$ below the hatched area cause a non-zero change in output. The hatched area equals $\mathbb{E}[dX(\varepsilon)]$, the expected size of a change for randomly drawn $\omega$.

For the full user-provided stochastic program, we are interested in computing $\frac{d\mathbb{E}[X(p)]}{dp}$, which is related to $\mathbb{E}[dX(\varepsilon)]$ by

$$\frac{d\mathbb{E}[X(p)]}{dp} = \lim_{\varepsilon \to 0} \frac{\mathbb{E}[dX(\varepsilon)]}{\varepsilon}. \tag{2.2}$$

We thus expect $dX(\varepsilon)$ to have "infinitesimal" $\mathcal{O}(\varepsilon)$ expectation as $\varepsilon$ approaches 0. Let us consider the form of $dX(\varepsilon)$ for two elementary programs: a program returning a sample from the continuous Exponential distribution $\text{Exp}(p)$ with scale $p$ and a program returning a sample from the discrete Bernoulli distribution $\text{Ber}(p)$ with success probability $p$. In both cases, the program is parameterized via the inversion method [31] over the sample space $\Omega = [0, 1]$, which chooses $\mathbb{P}$ to be uniform over $[0, 1]$ and $X(p)(\omega)$ to be non-decreasing in $\omega$. For $X(p) \sim \text{Exp}(p)$, $X(p)(\omega) = -p \log(1 - \omega)$ is differentiable in $p$. Thus, the range of $dX(\varepsilon)$ is $\mathcal{O}(\varepsilon)$ for any fixed $\omega$, as illustrated in Fig. 2a.

In contrast, consider $X(p) \sim \text{Ber}(p)$, with $X(p)(\omega) = \mathbf{1}_{[1-p,1]}(\omega)$. As shown in Fig. 2b, as $\varepsilon \to 0$ the output of $X(p)$ at a random $\omega \in \Omega$ almost surely does not change in $X(p + \varepsilon)$. Specifically, $dX(\varepsilon)$ is 0 with probability $1 - \varepsilon \approx 1$, and assumes the value 1 with probability $\varepsilon$ (Fig. 2c). This is the fundamental challenge presented by discrete randomness: a $\mathcal{O}(\varepsilon)$ change in the input turns into a *finite* perturbation to the output, where "finite" means that it is non-vanishing as $\varepsilon \to 0$. The perturbation occurs with "infinitesimal" $\mathcal{O}(\varepsilon)$ probability, contributing to the $\mathcal{O}(\varepsilon)$ expectation of $dX(\varepsilon)$.

## 2.2 Coupling to the primal

To produce low-variance estimates of the derivative of $X(p)$, it is important that the primal and derivative computations are coupled by a shared source of randomness. To see this, let us understand what happens when they are entirely uncoupled. A black-box finite difference approach with step size $\varepsilon$ would independently sample $\omega_1$ and $\omega_2$ from $\Omega$ (Fig. 2a, red), computing the derivative estimate $(X(p + \varepsilon)(\omega_2) - X(p)(\omega_1)) / \varepsilon$. However, since the samples are independent, the variance of the estimator is of order $1/\varepsilon^2$, so we are forced to pick a finite $\varepsilon$ to balance a bias-variance tradeoff.

This motivates using the same random sample for the primal and derivative computations (Fig. 2a, blue). For continuous randomness, taking the limit of this approach as $\varepsilon \to 0$ leads to the widely-used pathwise gradient estimator $\delta$, given as the almost sure limit of $dX(\varepsilon)/\varepsilon$ as $\varepsilon \to 0$ (i.e. the pointwise derivative of $X(p)$ at each fixed $\omega$), so that

$$\frac{d\mathbb{E}[X(p)]}{dp} = \lim_{\varepsilon \to 0} \frac{\mathbb{E}[dX(\varepsilon)]}{\varepsilon} \stackrel{?}{=} \mathbb{E}\left[\lim_{\varepsilon \to 0} \frac{dX(\varepsilon)}{\varepsilon}\right] = \mathbb{E}[\delta]. \tag{2.3}$$

But, considering a simple Bernoulli variable $X(p) \sim \text{Ber}(p)$ as in Fig. 2c, we see how this approach is ill-suited for the discrete case! As $\varepsilon$ approaches 0, the differential $dX(\varepsilon)$ is non-zero with

infinitesimal $\mathcal{O}(\varepsilon)$ probability. This means that $\delta$ is almost surely 0, while the true derivative of $\mathbb{E}[X(p)] = p$ is 1. The finite perturbation is neglected.

Thus, the pathwise gradient estimator needs to be modified to handle discrete randomness. The issue with the interchange of limit and expectation in (2.3) is that $\mathrm{d}X(\varepsilon)/\varepsilon$ is unbounded in $\varepsilon$ in the presence of a finite perturbation. This motivates explicitly considering the event of a large jump in $\mathrm{d}X(\varepsilon)$, as characterized by the event $A_B(\varepsilon) = \{|\mathrm{d}X(\varepsilon)| > B|\varepsilon|\}$ for a chosen random bound $B > |\delta|$. The event $A_B(\varepsilon)$ has $\mathcal{O}(\varepsilon)$ probability, which is vanishingly small, but its contribution to the derivative estimate cannot be neglected because it contains finite perturbations. This motivates sampling from this part of the probability space *separately*. Formally, we introduce a random weight $w \in \mathbb{R}$ and alternate value $Y \in E$ that characterize the sensitivity of $X(p)$ when the probability space is *restricted* to $A_B(\varepsilon)$ [as given by the r.h.s. of (2.4) below], forming the "stochastic derivative":

**Definition 2.2** (Stochastic derivative). Suppose $X(p) \in E$ is a stochastic program with index set $I$ a closed real interval. We say that the triple of random variables $(\delta, w, Y)$, with $w \in \mathbb{R}$ and $Y \in E$, is a right (left) *stochastic derivative* of $X$ at the input $p \in I$ if $\mathrm{d}X(\varepsilon)/\varepsilon \to \delta$ almost surely as $\varepsilon \to 0$, and there is an integrable (i.e. of bounded expectation) random variable $B > |\delta|$ such that for all bounded functions $f \colon E \to \mathbb{R}$ with bounded derivative it holds almost surely that

$$\mathbb{E}\left[ w\left(f(Y) - f(X(p))\right) \mid X(p) \right] = \lim_{\varepsilon \to 0^{+/-}} \mathbb{E}\left[ \frac{f(X(p+\varepsilon)) - f(X(p))}{\varepsilon} \mathbf{1}_{A_B(\varepsilon)} \,\middle|\, X(p) \right], \quad (2.4)$$

with limit taken from above (below), where $\mathbb{P}\left(A_B(\varepsilon) \mid X(p)\right)/\varepsilon$ is dominated by an integrable random variable for all $\varepsilon > 0$ ($\varepsilon < 0$).

A stochastic derivative may be collapsed into an unbiased estimator of the derivative of $\mathbb{E}[X(p)]$.

**Proposition 2.3** (Unbiasedness). *If $(\delta, w, Y)$ is a stochastic derivative of $X(p)$ at $p$, it holds that*

$$\frac{\mathrm{d}\mathbb{E}\left[X(p)\right]}{\mathrm{d}p} = \mathbb{E}[\delta + w\left(Y - X(p)\right)]. \quad (2.5)$$

*Proof sketch.* With $f$ as identity, by Definition 2.2 the sensitivity of $X(p)$ over $A_B(\varepsilon)$ is given by $w(Y - X(p))$. Given the complement event $A_B^c(\varepsilon)$, it holds that $|\mathrm{d}X(\varepsilon)/\varepsilon| \le B$: a dominated convergence argument shows that the sensitivity of $X(p)$ is then given by its almost-sure derivative $\delta$. $\square$

Theorem 2.4 shows the existence of the stochastic derivative for a stochastic program, subject to technical assumptions given in our formal treatment (Appendix B).

**Theorem 2.4** (Existence, simplified). *Given a sufficiently regular stochastic program $X(p)$ with index set $I$ a closed interval $[a, b] \subset \mathbb{R}$, there exists a right stochastic derivative $(\delta, w_R, Y_R)$ with $w_R \ge 0$ at any $p \in [a, b)$ and a left stochastic derivative $(\delta, w_L, Y_L)$ with $w_L \le 0$ at any $p \in (a, b]$.*

*Proof sketch.* The proof is by construction: at a high level, $w$ is the derivative of the probability of a large jump, while $Y$ follows the distribution of the possible jumps, conditional on a jump happening. Specifically, $Y$ has distribution given as the limit as $\varepsilon \to 0$ of the conditional distribution of $X(p+\varepsilon) = X(p) + \mathrm{d}X(\varepsilon)$ given the event $A_B(\varepsilon)$ and the outcome of $X(p)$. The weight $w$ is given as the derivative w.r.t. $\varepsilon$ of the probability $\mathbb{P}\left(A_B(\varepsilon) \mid X(p)\right)$ that $\mathrm{d}X(\varepsilon)$ jumps by a non-infinitesimal amount, conditional on $X(p)$. Essentially, since $\mathbb{P}\left(A_B(\varepsilon) \mid X(p)\right) \approx w\varepsilon$, multiplying by $w$ bridges the gap between *conditioning* on $A_B(\varepsilon)$ [recall $Y$ is constructed conditional on $A_B(\varepsilon)$] and simply restricting the probability space to $A_B(\varepsilon)$ [i.e. multiplying by $\mathbf{1}_{A_B(\varepsilon)}$ as in (2.4)]. $\square$

**Example 2.5** (Right stochastic derivative of Bernoulli variable). Suppose $X(p) \sim \mathrm{Ber}(p)$, parameterized via the inversion method, and take $\varepsilon > 0$. As shown in Fig. 2c, $\mathrm{d}X(\varepsilon)$ is parameterized as,

$$\mathrm{d}X(\varepsilon)(\omega) = \begin{cases} 1 & \text{if } 1 - p - \varepsilon \le \omega < 1 - p, \\ 0 & \text{otherwise.} \end{cases} \quad (2.6)$$

Given the event $X(p) = 1$, we have that $\omega \ge 1 - p$ and $\mathrm{d}X(\varepsilon)$ is deterministically 0. On the other hand, given $X(p) = 0$, we have $\omega < 1 - p$, so with probability $\varepsilon/(1 - p)$ the differential $\mathrm{d}X(\varepsilon)$ assumes a value of 1, i.e. the Bernoulli variable flips from 0 to 1. Thus, we may construct a right

stochastic derivative $(0, w_R, Y_R)$ of $X(p)$ by letting $w_R = 1/(1-p), Y_R = 1$ conditionally on $X(p) = 0$ and $w_R = Y_R = 0$ conditionally on $X(p) = 1$. For a concrete example, with $p = 0.6$, we have $w_R = 2.5, Y_R = 1$ conditionally on $X(p) = 0$; `StochasticAD.jl` prints this as `"0 + (1 with probability 2.5ε)"` (as explained further in Section 3.1).

We give a number of examples of stochastic derivatives in Appendix A. It is of crucial importance that $w$ and $Y$ depend conditionally on the output of $X(p)$. In particular, given a primal evaluation $X(p) = x$, the forms of $w$ and $Y$ depend only on the distribution of the perturbed program on the set $\{\omega : X(p)(\omega) = x\} \subset \Omega$, which elegantly generalizes the coupling achieved by the pathwise gradient estimator $\delta$ at each fixed $\omega \in \Omega$. Intuitively, this coupling allows us to consider only the smallest possible perturbations to the program in the derivative computation (Fig. 1), and thereby achieve variance reduction without resorting to continuous relaxations. For example, for a binomial variable $X(p) \sim \mathrm{Bin}(n, p)$ parameterized via the inversion method (a natural parameterization for maximizing coupling), it holds that $Y \in \{X(p) - 1, X(p) + 1\}$. We show in Example A.1 that our gradient estimator for such a binomial variable has variance of order $n$, whereas the score function estimator has variance of order $n^3$, justifying this intuition.

## 2.3 Composition of stochastic derivatives

Proposition 2.3 shows that the derivative estimate produced by stochastic derivatives is correct in expectation. But this is insufficient to ensure *composition*, i.e. a stochastic derivative "chain rule". While Definition 2.2 requires composition through deterministic test functions $f$ to enforce a sufficiently strict definition, Theorem 2.6 provides a general-purpose composition result through any program which has a stochastic derivative, as well as multidimensional programs with directional stochastic derivatives (i.e. stochastic derivatives of directional perturbations of the program).

**Theorem 2.6** (Chain rule, simplified)**.** *Consider independent stochastic programs $X_1$ and $X_2$ and their composition $X_2 \circ X_1$. Suppose that $X_1$ has a right (left) stochastic derivative at $p \in \mathbb{R}$ given by $(\delta_1, w_1, Y_1)$, and $X_2$ has a right stochastic derivative $(\delta_2, w_2, Y_2)$ in the direction $\widehat{\delta}_1 = \delta_1/|\delta_1|$ given conditionally on its input $X_1(p)$. Then, under regularity and integrability assumptions, the stacked program $[X_1; X_2 \circ X_1]$ has a right (left) stochastic derivative $(\delta, w, Y)$ at $p$ where $\delta = [\delta_1; |\delta_1|\delta_2]$,*

$$Y \quad = \quad \begin{cases} [Y_1; X_2(Y_1)] & \textit{with probability} \quad \dfrac{w_1}{w_1 + |\delta_1|w_2}, \\ [X_1(p); Y_2] & \textit{with probability} \quad \dfrac{|\delta_1|w_2}{w_1 + |\delta_1|w_2}, \end{cases} \tag{2.7}$$

*and $w = w_1 + |\delta_1|w_2$.*

*Proof sketch.* Let $A_1(\varepsilon)$ be the event of a jump in $X_1$ and $A_2(\varepsilon)$ be the event of a jump in $X_2$ when its input is perturbed by $\varepsilon\delta_1$. Given $A_1^c(\varepsilon) \cap A_2^c(\varepsilon)$ (a jump in neither), a dominated convergence argument implies that $X_2 \circ X_1$ does not jump either. Given $A_1(\varepsilon)$, Definition 2.2 yields that the sensitivity of $X_2 \circ X_1$ is described by alternate values $Y_1$ with weight $w_1$, while given $A_2(\varepsilon)$ the sensitivity of $X_2 \circ X_1$ is described by alternate values $Y_2$ with weight $|\delta_1|w_2$. We may then form $w$ as the sum of these weights and $Y$ as a weighted distribution over the two cases. Crucially, we may neglect the sensitivity of $X_2 \circ X_1$ given $A_1(\varepsilon) \cap A_2(\varepsilon)$ (a jump in both), as this event has probability $\mathcal{O}(\varepsilon^2)$. This prevents a combinatorial explosion in the complexity of $Y$. $\qquad\square$

## 2.4 Smoothed stochastic derivatives

The reparameterization trick neglects finite perturbations, while stochastic derivatives precisely capture all possible finite perturbations. There exists a middle ground between these methods: one may take a conditional expectation on $X(p)$ so that finite perturbations have been "smoothed" into infinitesimal ones. (E.g. `"0 + (1 with probability 2.5ε)"` becomes `"0 + 2.5ε"`.)

**Definition 2.7** (Smoothed stochastic derivative)**.** For a stochastic program $X(p)$ with a right (left) stochastic derivative $(\delta, w, Y)$ at input $p$, a right (left) smoothed stochastic derivative $\widetilde{\delta}$ of $X$ at input $p$ is given as

$$\widetilde{\delta} = \mathbb{E}\left[\delta + w(Y - X(p)) \mid X(p)\right]. \tag{2.8}$$

```julia
1 struct StochasticTriple
2     value # primal evaluation
3     δ # "infinitesimal" component
4     Δs # component of discrete change
5         # with "infinitesimal"
6         # probability
7 end
```

```julia
1 using Distributions
2 function X(p)   p = 0.6 + ε
3     a = p^2   0.36 + 1.2ε
4     b = rand(Binomial(10, p))
5           6 + (1 with probability 10.0ε)
6     c = 2 * b + 3 * rand(Bernoulli(p))
7           12 + (3 with probability 12.5ε)
8     return a * c * rand(Normal(b, a))
9 end
```

```julia
1 julia> using StochasticAD
2 julia> st = stochastic_triple(X, 0.6) # sample a single stochastic triple at p = 0.6
3 27.11 + 94.32ε + (6.78 with probability 12.5ε)
4 julia> derivative_contribution(st) # which produces a single derivative estimate...
5 179.04
6 julia> samples = [derivative_estimate(X, 0.6) for i in 1:1000] # take many estimates!
7 julia> println("d/dp of E[X(p)]: $(mean(samples)) ± $(std(samples) / sqrt(1000))")
8 d/dp of E[X(p)]: 204.63 ± 1.25
```

Figure 3: **Left:** Stochastic triple structure (simplified). **Right:** A toy program $X(p)$, using discrete distributions $\text{Bin}(n, p)$, $\text{Ber}(p)$, and the continuous normal distribution $\mathcal{N}(\mu, \sigma)$; the used stochastic derivatives are given in Appendix A. Highlights show intermediate values during a single derivative estimate. **Bottom:** Differentiating $\mathbb{E}[X(p)]$; printout float precision reduced for clarity.

Smoothed stochastic derivatives easily permit reverse-mode AD instead of forward-mode, as they have the same form as the usual derivative. However, they enjoy more limited composition properties: they propagate exactly through differentiable functions $f$ that are linear over the conditional distribution of $Y$ given $X(p)$ via the standard chain rule, as we prove in Appendix B.6. Due to the coupling of $Y$ and $X(p)$, this is a much weaker requirement than global linearity, which can lead to low-bias estimates: for example, for the program $X(p) = \text{Geo}(p)^3$ smoothed stochastic derivatives give a derivative estimate with $< 0.5\%$ bias at $p = 0.01$, even though the cube function is highly non-linear on the inter-quartile range $[28, 137]$ of $\text{Geo}(p)$. Smoothed stochastic derivatives recover the widely-used straight-through gradient estimator [6] as a special case, as we work out in Example A.8.

## 3 Automatic differentiation of stochastic programs

We develop `StochasticAD.jl`, a prototype package for stochastic AD based on our theory of stochastic derivatives. As discussed in Section 2.4, *smoothed* stochastic derivatives obey the usual chain rule, and thus can be used with existing AD infrastructure by supplying custom rules for discrete random constructs, and we do so for a particle filter in Section 3.4. However, performing automatic differentiation with unsmoothed stochastic derivatives, which are unbiased in all cases, requires new innovation. We develop a novel computational object called a *stochastic triple*, introduced in Section 3.1 and showcased in Section 3.2 and Section 3.3.

### 3.1 Educational toy example of stochastic triples

Forward-mode AD is often implemented with *dual numbers* [2], which pair the primal evaluation of a deterministic function $f(p)$ with its derivative $\frac{\mathrm{d}}{\mathrm{d}p}f(p)$. Dual numbers can be propagated through a program using the chain rule. A useful alternative perspective of dual numbers is that they propagate an "infinitesimal" perturbation $\varepsilon$ to the input through the program, where $\frac{\mathrm{d}}{\mathrm{d}p}f(p)$ is the coefficient of the "dual" element $\varepsilon$. Stochastic triples generalize dual numbers by including a third component $\Delta$s to describe finite perturbations with infinitesimal *probability* (Fig. 3, left).

In Fig. 3, we consider a toy program $X(p)$ including discrete randomness. We are interested in the derivative of $\mathbb{E}[X(p)]$ at $p = 0.6$, and hence we provide the stochastic triple printed as `"0.6 + ε"` as input. First, the triple is squared and becomes `"0.36 + 1.2ε"`: this is the familiar way that dual numbers propagate, via the chain rule. But what happens when the triple is propagated

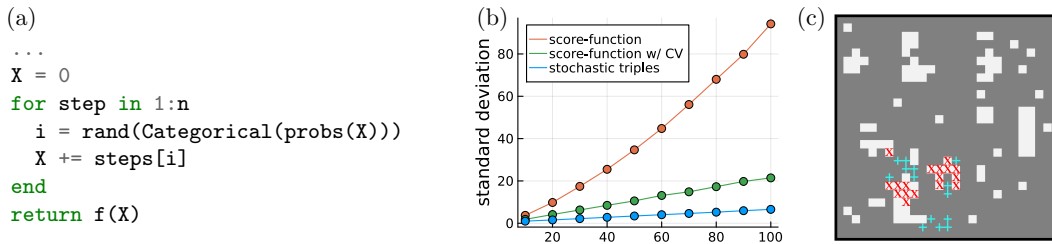

(a)
```
...
X = 0
for step in 1:n
  i = rand(Categorical(probs(X)))
  X += steps[i]
end
return f(X)
```

(b)

(c)

Figure 4: Automatic differentiation of discrete-time Markov processes. **(a)** Code snipped for a 1D random walk, which can be automatically differentiated by `StochasticAD.jl`; `probs(X)` gives the transition probabilities at value `X` and `steps[i]` gives the step size for the `i`th transition. **(b)** The variance of unbiased gradient estimates of the random walk program using stochastic triples and the score function, which is applied both without a control variate (CV) and with a pre-computed batch-average CV. **(c)** The final board for one run of the stochastic Game of Life with $N = 25$ and $T = 10$, where the "+" signs represent additional living cells (white) in the stochastic alternative path, and the "X" signs represent additional dead cells (grey).

through the discrete and random Binomial variable? The resultant stochastic triple `"6 + (1 with probability 10.0ε)"` is an integer with a component of *discrete* change, reflecting an infinitesimal probability of one more success in the Binomial $\mathrm{Bin}(10, 0.6)$. We can in fact understand why the probability is `"10ε"` by representing the Binomial as the sum of 10 Bernoulli variables, each with probability $0.6$. Since 4 of the Bernoulli's have an output of 0, they each have a probability `"2.5ε"` of switching to 1 (recall Example 2.5), and thus there is in total a `"10ε"` probability that the output of the Binomial increases by 1 (rigorously, we have applied Theorem 2.6). Formally, we can interpret a printout `"x + δε + (Δ with probability wε)"` as follows: `x` is a sample of the random variable $X(p)$ describing the primal evaluation, and `δ`, `w`, and `x+Δ` are samples of the components $\delta$, $w$, and $Y$, respectively, of the stochastic derivative of $X$ at $p$.

Using our chain rule for stochastic derivatives (Theorem 2.6), we write rules for propagating stochastic triples through functions via operator overloading [32], exploiting Julia's multiple dispatch feature [33]. When multiple discrete changes are possible, we pick one probabilistically: we call this strategy *pruning* (recall Fig. 1) and show its unbiasedness in Appendix B.5. For example, in line 6 of Fig. 3, right, we probabilistically choose between the perturbation to the Binomial and the perturbation to the Bernoulli, in this case picking the latter. To handle causal relationships between perturbations as in the first case of Eq. (2.7), we associate each perturbation with a tag to avoid erroneously pruning between two perturbations that occur simultaneously. Thus, stochastic triples can efficiently propagate through the full toy function written in Fig. 3. The function `derivative_estimate` creates a stochastic triple, propagates it, and collapses it into the derivative estimate `δ + wΔ`, forming an unbiased estimate of the derivative via Proposition 2.3 (Fig. 3, bottom).

### 3.2 Inhomogeneous random walk

We consider a Markovian random walk $x_0, \ldots, x_n$ on $\mathbb{Z}_{\geq 0}$, with transition behavior dependent on a parameter $p$ as follows,

$$x_n = \begin{cases} x_{n-1} + 1 \text{ with probability } \exp\left(-\frac{x_{n-1}}{p}\right) \\ x_{n-1} - 1 \text{ with probability } 1 - \exp\left(-\frac{x_{n-1}}{p}\right) \end{cases}, \quad x_0 = 0. \quad (3.1)$$

We consider a program that stochastically simulates this walk and applies an arbitrary non-linear function $f$ to the output $x_n$. In practice, $f$ may represent a loss or a likelihood estimate; in this toy setting, we take $f(x) = x^2$. We are interested in studying the asymptotic behavior of the variance of our automatically derived gradient estimator, and so set $p = n$ so that the transition function varies appreciably over the range of the walk for all $n$. We find that the stochastic triple estimator has asymptotically lower variance than the score function estimator (Fig. 4b). Crucially, stochastic triples achieve this variance reduction while remaining entirely in discrete space, automatically producing a gradient estimate that is provably unbiased.

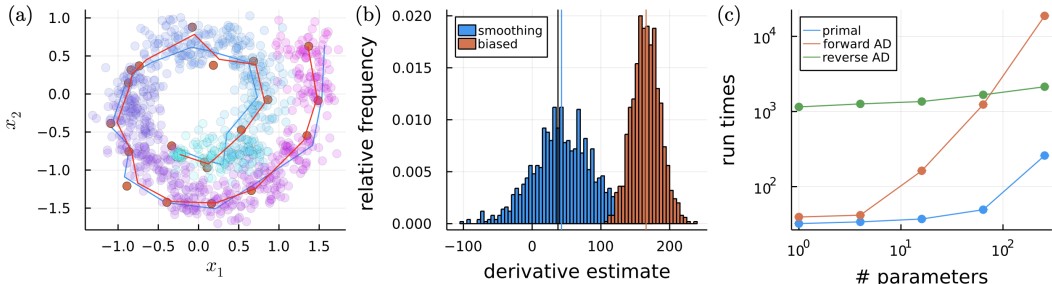

Figure 5: Unbiased differentiable particle sampler using smoothed stochastic derivatives. **(a)** Latent path (blue) and observations (orange) for a particular realization of the process given by the hidden Markov model (see Appendix C for details) in $d = 2$ dimensions, including the recursively applied Kalman filter estimator (red) and the particles of the particle filter (going from blue at $n = 1$ to pink at $n = 20$). **(b)** The value of the derivative of $\log \mathcal{L}$ with respect to the first parameter for $d = 2$ calculated by differentiating the Kalman filter is marked by a black line. The blue distribution corresponds to the derivative computed with smoothed stochastic derivatives. The orange distribution represents the biased derivative approach, where the resampling step is not differentiated. The means of the two distributions are highlighted by a line in the respective color. In both cases, we use 1000 samples. **(c)** Averaged run times in $\mathrm{ms}$ comparing forward- and reverse-mode AD for increasing numbers of parameters.

## 3.3 Stochastic Game of Life

For our next example, we differentiate a stochastic version of John Conway's Game of Life, played on a two-dimensional board. In the traditional Game of Life, a dead cell becomes alive when 3 of its neighbors are alive, while a living cell survives when 2 or 3 of its neighbors are alive. In our stochastic version, each of these events instead has probability $95\%$, while their complementary events have probability $5\%$. Such types of discrete stochastic programs arise in many applications. For example, mixing machine learning with agent-based models found in epidemiology and sociological contexts [34, 35] or rule-based models and discrete stochastic (Gillespie) simulations in systems biology [36, 37] requires similar program constructs.

Consider a program that populates each cell of an $N \times N$ board with probability $p$, runs the stochastic Game of Life for $T$ time steps, and counts the number of living cells $n_{\text{living}}$. We perform a sensitivity analysis of the final living population with respect to the initial living population, i.e. differentiate the expectation of $n_{\text{living}}$ with respect to $p$. Stochastic triples propagate fully through the program, leading to an unbiased estimate of the derivative, as we verify with black-box finite differences. An example final board is depicted in Fig. 4c, along with the difference to the alternative final board chosen by pruning. This is a high-dimensional example (the state space has dimension $N^2 = 625$) with fundamentally discrete structure, providing support for the algorithmic correctness and generality of stochastic triples. In particular, the program cannot directly be continuously relaxed since it includes array indexing; in the approach of stochastic triples, integer-valued quantities stay integers.

## 3.4 Particle filter

As a final example, we consider a hidden Markov model with random latent states $X_1, \ldots, X_n$, observations $y_1 \sim Y_1, \ldots, y_n \sim Y_n$, and parameters $\theta$. The likelihood $\mathcal{L} = p(y_1, \ldots, y_n \mid \theta)$ is in general not tractable, but a particle filter can be used to compute an estimate of $\mathcal{L}$. Here, we assume familiarity with particle filters [38, 39] but provide a short exposition with focus on the resampling step and experimental details in Appendix C. We assume the latent states to be continuous random, but discrete randomness enters through the resampling step.

To differentiate the particle filter resampling step, we provide a stochastic derivative formulation of the returned particles and importance weights (Appendix C). Importantly, $\mathcal{L}$ can be expressed by the sum of the weights at the last step, and the weight assigned to each particle in the resampling steps

is used in a *purely linear* way. Smoothed stochastic derivatives permit *unbiased* reverse-mode AD in this case. Our formulated approach, though derived very differently, is equivalent to the particle filter AD scheme developed in [7] implementing the first estimator derived in [40], as we show in Appendix C. Our particular choice of system allows the calculation of a ground-truth gradient of $\mathcal{L}$ by differentiating the Kalman filter algorithm [41]. Fig. 5a visualizes the latent process and observations, and the Kalman and particle filter trajectories. Our estimator agrees with the Kalman filter derivative, unlike biased estimators [42, 43, 44] that neglect the contribution of the resampling step or perform it with entropy-regularized optimal transport (Fig. 5b). For our program, we observe reverse-mode AD to perform better for more than $\approx 100$ parameters (Fig. 5c). However, we find that the variance increases more rapidly with the number of steps and dimension as compared to biased estimators, suggesting that there is room for improvement in the coupling approach [7, 40].

## 4 Limitations and outlook

We have presented a method for unbiased AD of programs with discrete randomness, which, we have argued, is a natural generalization of pathwise gradient estimators based on the reparameterization trick to the discrete case. However, more work will need to be done to turn our software demonstration `StochasticAD.jl` into an AD system capable of handling the full complexity of applications such as machine learning. A useful improvement would be better support for discrete constructs such as `if` statements with discrete random input; currently, such branches need to be rewritten using array indexing, which is supported. Further, an interesting direction for future work is to automatically handle functions which deterministically turn a continuous random quantity into a discrete one, such as an inequality comparison `X(p) > 0` or a Bernoulli variable implemented implicitly as `rand() < p` and also to handle constructs such as while loops based on such functions. It would also be interesting to explore synergies of our approach with ADEV [45], a Haskell-based framework for provably correct stochastic AD developed concurrently with our work, which formulates gradient estimation strategies using Haskell's continuation passing style.

We also expect future work to focus on further variance reduction. Our method's variance depends on the degree to which the primal and derivative computations can be *coupled*: while we present a natural method of coupling for a number of elementary stochastic programs and their compositions, the design space is rich when it comes to challenging examples such as the Game of Life or the resampling step of a particle filter. (This design space is reflected in the fact that the form of the stochastic derivative depends not only on a program's probability distribution, but also on the way it is parameterized.) Furthermore, the pruning operation can introduce additional variance: for example, the derivative estimate automatically produced for the program $B_1 + 2B_2$ with i.i.d. $B_i \sim \text{Bin}(n, p)$ has variance $\mathcal{O}(n^2)$ due to the pruning between the +1 and +2 perturbations, even though stochastic derivatives give gradient estimators with variance $\mathcal{O}(n)$ for each of $B_1$ and $B_2$. Smoothing does not face this issue but accrues a bias through non-linear functions. Pruning and smoothing may be thought of as the simplest ways to construct an AD algorithm from stochastic derivatives, lying on opposite ends of the design space: we anticipate future work to address their suboptimalities and ideally form unbiased estimators for discrete random programs that fully close the variance gap to their continuous counterparts [14, 46]. Finally, going beyond smoothing for reverse-mode AD, and ideally achieving unbiasedness while remaining coupled to the primal, is an important open problem for large-scale applications.

## Acknowledgments and disclosure of funding

We thank Alan Edelman, Guillaume Dalle, and the anonymous reviewers for their feedback, Emile van Krieken for helpful discussions regarding composing gradient estimators, and Simeon Schaub for help with the package. We acknowledge the MIT SuperCloud and Lincoln Laboratory Supercomputing Center for providing HPC resources that have contributed to the research results reported within this paper. This material is based upon work supported by the National Science Foundation OAC-1835443, SII-2029670, ECCS-2029670, OAC-2103804, and PHY-2021825; the Advanced Research Projects Agency-Energy DE-AR0001211 and DE-AR0001222; the Defense Advanced Research Projects Agency (DARPA) HR00112290091; the United States Artificial Intelligence Accelerator FA8750-19-2-1000; the Chalmers AI Research Centre; and the Swiss National Science Foundation 51NF40-185902. The views and opinions of authors expressed herein do not necessarily state or reflect those of the United States Government or any agency thereof.

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
