# OpenReview forum: "Automatic Differentiation of Programs with Discrete Randomness"
_NeurIPS.cc/2022/Conference — NeurIPS 2022 Accept_

### Official Review · Reviewer_kqcs · 2022-07-11

**Rating:** 6
**Confidence:** 3
**Soundness:** 3 good
**Presentation:** 2 fair
**Contribution:** 4 excellent

**Summary:**

The paper provides a new method to compute the gradient of the expected output of a stochastic computation graph (SCG) w.r.t one of the input parameters of the graph even when the graph contains discrete randomness.

In a well known method for computing gradients of stochastic computation graphs when they contain only continuous valued variables all the stochastic variables are reparameterized which allows for the gradient of the expectation to be sampled for any parameter by simply propagating the primal value of the graph variables as well as the forward gradients. This paper extends this well-known method by additionally propagating an alternate value for one of the discrete stochastic nodes in the graph (plus the alternate values of all the children of that flipped node) as well as the gradient of the probability of that alternate value.

This method has been shown to greatly reduce the variance of the estimated gradient of the expectation as opposed to standard methods such as the score function estimator which is typically used to estimate such gradients for graphs with discrete randomness.


**Questions:**

In lines 311-312 the authors say that the method is only applicable when control flow is independent of random parameters. What does this mean exactly? Does this mean that if the SCG has if-statements and it can take completely different paths based on random values then the method doesn't work. Why?

In Figure 1 why does it show 3 alternate paths when pruning has been defined to only allow one alternate path?

In definition 2.2, it seems to me that `w` is the gradient of the probability of the output of the SCG flipping from X to Y (where the gradient is w.r.t. p). Is this an accurate charaterization?

In lines 151-159 there seems to be some confusion as to whether w is a vector or a scalar based on the description. Could you please give a clearer description?

In line 181 where does the $\delta$ come from? The statement of the proposition has $\delta_1$ and $\delta_2$? This proof is also hard to follow in the supplement. Could you provide the key points of the proof here? I would give a higher rating and confidence if I can convince myself that these composition propositions are correct.

The description of smoothed stochastic duals is very vague. Could you give a more mathematical definition?

In Figure 5 what does `rand(Categorical(probs[X]))` mean?

# Minor points

- Line 142 "program with parameterized" seems to be missing something?

- Figure 6 left figure takes very long to render. Please consider alternate image formats.

- Line 242 typo: "filterunbiasedness"

- In equation 4.1 can x be negative? If yes then the probabilities don't make sense.



**Limitations:**

No known negative societal impact.

**Strengths And Weaknesses:**

The method is clearly original. There are a number of papers in the literature which attempt to solve this particular problem (estimate gradients of SCG with discrete random variables) such as reinforce, concrete, score function estimator, path wise gradients etc. But they are all rather different from the current paper and those methods all suffer from some form of high variance issues that are explicitly addressed in this paper.

This paper's key contribution is the realization that simply estimating gradients along multiple independent paths may introduce excessive variance in the estimate and it is better to look for correlated paths.

The work is very signification since estimating gradients in SCGs are an important problem in various branches of AI in particular Reinforcement Learning and Bayesian Inference. There are a number of papers routinely published every year in NeurIPS which make some reference or the other to this problem and use one of the aforementioned methods. So, an improved solution to this problem is hugely impactful.

The weakness of the approach is that it is only defined for forward mode gradients and the authors only spend any effort to describe the gradient computation for a single input. In most machine learning applications reverse mode gradients are preferable to compute since we have one loss function which we want to differentiate w.r.t. multiple parameters.

The quality of the technical writing is somewhat patchy though and that is why I hesitate to give a higher rating than would be deserved for work that has provided such an interesting breakthrough.

The experiments have been selectively picked for examples with a single parameter, but that seems a bit contrived. There are a number of reinforcement learning papers in the literature which rely on SCG differentiation. Using one of them in this paper would have been more impactful than the game of life example. The resampling example on the other hand was interesting as that is a common use case.

---

> ### Author Response · Authors · 2022-08-02
> **Response to Reviewer kqcs**
>
> We thank the reviewer for their insightful feedback and their positive comments regarding the novelty and impact of our approach. To address the reviewer’s feedback, we have revised Sections 2 and 3 to incorporate more mathematical, rigorous statements to improve the overall technical soundness of the paper. We have also reworked the key proofs in the supplement, filling in details to make them more thorough and also more accessible.
> > The quality of the technical writing is somewhat patchy though and that is why I hesitate to give a higher rating than would be deserved for work that has provided such an interesting breakthrough.
>
> We have spent significant effort revising the technical clarity of the paper based on your feedback. We hope that the bulleted changes we listed in our general response, as well as our separate response to your technical questions, address your concerns regarding clarity.
>
> > The weakness of the approach is that it is only defined for forward mode gradients and the authors only spend any effort to describe the gradient computation for a single input. In most machine learning applications reverse mode gradients are preferable to compute since we have one loss function which we want to differentiate w.r.t. multiple parameters.
>
> We agree with the reviewer that for highly parameterized models, reverse-mode AD will be desirable and that deriving a reverse-mode AD algorithm based on unsmoothed stochastic derivatives is an exciting open problem that will be the subject of future research. However, we want to highlight that smoothed stochastic derivatives have allowed us to compute an unbiased derivative estimate of the particle sampler in reverse-mode AD, with respect to a large number of inputs. We clarified Section 2.4 in this respect.
>
> > The experiments have been selectively picked for examples with a single parameter, but that seems a bit contrived. There are a number of reinforcement learning papers in the literature which rely on SCG differentiation. Using one of them in this paper would have been more impactful than the game of life example. The resampling example on the other hand was interesting as that is a common use case.
>
> Our intent with the game of life was to highlight the composability of stochastic triples by considering a program written at the language level with discrete constructs such as array indexing that may not easily fit into the SCG framework. Furthermore, arbitrary compositionality is the key ingredient (this could correspond to a very large and structured SCG). In this sense, this is a particularly challenging example with the potential to extend differentiable programming and ML to new domains, as we note at the end of our general response.
>
> > In lines 311-312 the authors say that the method is only applicable when control flow is independent of random parameters. What does this mean exactly? Does this mean that if the SCG has if-statements and it can take completely different paths based on random values then the method doesn't work. Why?
>
> We have expanded on the topic of control flow in the “limitations and outlook” section. Currently, `if` statements dependent on discrete random input fit into our theoretical framework. But these are not yet supported on the software side; in the game of life example, we use array indexing as a workaround, which is supported. Note that if the structure of the computation completely changes depending on an `if` statement, then no coupling can be achieved between these two paths; however, this is still an important feature to support, as we would still be able to couple other parts of the computation.
> > In Figure 1 why does it show 3 alternate paths when pruning has been defined to only allow one alternate path?
> The intention was to also show the alternative paths that were not picked by pruning (the pruning strategy may follow these paths for some time before switching to the one it ultimately chooses). We have revised the figure to highlight the path chosen by pruning.
> > Figure 6 left figure takes very long to render. Please consider alternate image formats.
>
> We have changed the figure from pdf to png to avoid the rendering problems.
> > Line 142 "program with parameterized" seems to be missing something?
> > Line 242 typo: "filterunbiasedness"
>
> Thanks for pointing these out, we have fixed the typos in the revised version.

---

> > ### Author Response · Authors · 2022-08-02
> > **Addressing technical questions of Reviewer kqcs**
> >
> > > In definition 2.2, it seems to me that w is the gradient of the probability of the output of the SCG flipping from X to Y (where the gradient is w.r.t. p). Is this an accurate characterization?
> >
> > That’s right. The only qualification is that the sign of the limit affects the form of $w$, as stochastic derivatives are in general asymmetric. We have included an explicit formula for $w$ in the main text itself, which we hope helps to clarify its definition.
> >
> > > In lines 151-159 there seems to be some confusion as to whether w is a vector or a scalar based on the description. Could you please give a clearer description?
> >
> > We have clarified that the weight function $w$ has real output. (Strictly speaking, $w$ is a real-valued random variable rather than a scalar. But given a particular primal evaluation, $w$ always assumes a scalar value.)
> >
> > > In line 181 where does the $\delta$ come from? The statement of the proposition has $\delta_1$ and $\delta_2$ ? This proof is also hard to follow in the supplement. Could you provide the key points of the proof here? I would give a higher rating and confidence if I can convince myself that these composition propositions are correct.
> >
> > We have corrected this typo in the main text. Furthermore, we have stated the exact same proposition in the main text as what we prove in the appendix to reduce ambiguity, and we have made the composition proof clearer and more precise so that it can be followed step by step.
> >
> > The key idea of the proof is to partition the sample space into four events, corresponding to the occurrence or non-occurence of a finite perturbation in $X_1$ and $X_2$. In the proof’s notation (following the revised version), these events are $A_1(\varepsilon) \cap \tilde{A}_2(\varepsilon)$, $A_1(\varepsilon) \cap \tilde{A}_2^c(\varepsilon)$, $A_1^c(\varepsilon) \cap \tilde{A}_2(\varepsilon)$, and $A_1^c(\varepsilon) \cap \tilde{A}_2^c(\varepsilon)$.
> > The event $A_1(\varepsilon) \cap \tilde{A}_2(\varepsilon)$ of finite perturbations in both variables has $\mathcal{O}(\varepsilon^2)$ probability, so its contribution to the sensitivity of $f \circ X_2 \circ X_1$ approaches 0. Now, given the events $A_1(\varepsilon) \cap \tilde{A}_2^c(\varepsilon)$ and $A_1^2(\varepsilon) \cap \tilde{A}_2(\varepsilon)$, we can use the constructions of $w_1,w_2,Y_1$ and $Y_2$ to figure out:
> > * what the stacked program $[X_1; X_2 \circ X_1]$ converges to in distribution given one of these events (e.g.  $[Y_1; X_2(Y_1)]$ for $A_1(\varepsilon) \cap \tilde{A}_2^c(\varepsilon)$)
> > * the derivative of the probability these events conditional on the primal.
> > This gives us precisely the setting of Assumption B.2 for each of these events, and hence analogous reasoning to Proposition B.3 lets us characterize the sensitivity of $f \circ [X_1; X_2 \circ X_1]$ given that one of the events occurs as $w \left(f(Y) - f( [X_1; X_2 \circ X_1])\right)$.
> >
> > Finally, we can apply an analogous argument to Proposition B.1 to conclude that the program is well-characterized by the reparameterization trick given the event $A_1^c(\varepsilon) \cap \tilde{A}_2^c(\varepsilon)$, which holds almost surely as $\varepsilon \to 0$. After characterizing the sensitivity of $f \circ X_2 \circ X_1$ on each of the four events, we can sum everything up to conclude the claim.
> >
> > > The description of smoothed stochastic duals is very vague. Could you give a more mathematical definition?
> >
> > We have given a formal definition of smoothed stochastic derivatives in the revised text, namely a random variable $\tilde{\delta}$ that has the same expectation as the stochastic derivative when conditioned over the primal output: $\mathbb{E}[\tilde{\delta} | X(p)] = \mathbb{E}[\delta + w(Y-X(p)) | X(p)]$. We have removed the description of smoothed stochastic duals from the main text for clarity, since dual numbers are simply another perspective on derivatives, and no compiler/software innovations are required to perform AD with smoothed stochastic derivatives.
> > > In Figure 5 what does `rand(Categorical(probs[X]))` mean?
> >
> > Here, `probs[X]` represents the probability vector giving all possible transitions for a walk at position `X`, as we have clarified in the figure caption. This way the code allows a walk that may have more than two possible transitions. The full line returns a random integer between 1 and the total number of possible steps, which is used as an index in the next line to select the step size.
> >
> > > In equation 4.1 can x be negative? If yes then the probabilities don't make sense.
> >
> > Since the probability of going right is 1 at $x = 0$, the states have domain $\mathbb{Z}_{\geq 0}$, which we have clarified in the revision.

---

### Official Review · Reviewer_x6td · 2022-07-11

**Rating:** 7
**Confidence:** 2
**Soundness:** 3 good
**Presentation:** 3 good
**Contribution:** 3 good

**Summary:**

This paper proposes a novel approach for both forward- and reverse-mode automatic differentiation of probabilistic programs with discrete RVs. Since the derivative $\frac{dX}{dp}$ of a program $X$ with respect to the parameters $p$ of a discrete distribution is not well-defined, the authors circumvent the problem by finding the derivative of the expectation of $X$, i.e. $\frac{d\mathbb{E}(X)}{dp}$.
The technique results in an unbiased estimator, exhibiting lower variance with respect to prior approaches.


**Questions:**


1- In 4.1, what value(s) of $p$ were used? Why not showing in Fig.5  (center) the standard deviation over multiple values of $p$?

2- In 4.2 the program is almost deterministic with $p=0.05$. What happens as $p$ approaches $0.5$?

3- Why not applying the score-function method to the tasks 4.2 and 4.3? It is also not clear to me why some approaches mentioned in the Related Work section were not considered (REBAR, RELAX, or the reparametrization trick with Gumbel-softmax).

4- In Fig.6 (center), why not reporting both forward and reverse mode AD, possibly using the same colors in Fig.6 (right)?

5- The paper mentions some additional complications when dealing with multivariate functions. Why not reporting experiments in that sense?

---

**Minor comments**

- Fig.1: I don't get it. $X'(p + \epsilon)$ with random number sequence
  z(2) (bottom) in the caption is $X(p + \Delta p)$ in the figure. What
  is the relation between $\Delta p$ and $\epsilon$?
---
"This variance issue can be alleviated if one could run the perturbed program with the same set of random numbers, i.e. directly compute $\mathbb{E}[X(p + \epsilon) − X(p)]$"

- Shoudn't it be 'solved' rather than 'alleviated'?
---
- Def. 2.2 "right (left) derivative" 'stochastic' is missing.
---
- When is a stochastic program "sufficiently regular"?
---

"It is convenient to assume probability densities [...]  and transition probability densities [...] all depending on the parameters $\theta$."

- Why? You mean from a notational perspective?
---

**Typos**

- "algorithm. enabling"
- "an weighted"
- "with parametrized by"
- "particle filterunbiasedness"






**Limitations:**

Mostly yes, although the empirical evaluation could be improved in this sense (see above).

**Strengths And Weaknesses:**

The problem considered in this work is a relevant one.
While the paper is generally well-written, I found it hardly accessible for nonexperts in AD, like myself.
I wish that the paper included some more examples, for instance a figure showing how these stochastic triples are propagated in a simple program. In this sense, Figures 3 and 4 are not that useful in my opinion. I can't really assess the soundness of the proposed approach, but I didn't spot any major issues. Similarly, I am not familiar with the existing literature and it is possible that I missed some relevant work. I think that the experimental evaluation could be improved (see questions below).

**Update after rebuttal** The authors addressed my points effectively and submitted a significantly improved version of the manuscript.

---

> ### Author Response · Authors · 2022-08-02
> **Response to Reviewer x6td**
>
> We thank the reviewer for the helpful feedback, and for assessing our work as relevant and generally well-written. We acknowledge that accessibility for non-AD-experts could be improved and are thankful for the reviewer’s detailed feedback: On the reviewer’s recommendation, we have included a more detailed code figure of how stochastic triples are propagated in a simple program, see Fig. 3 in our revised version, and Section 3.1 for the accompanying description.
>
> > I wish that the paper included some more examples, for instance a figure showing how these stochastic triples are propagated in a simple program.
>
> We have revised our code figure to provide a more extensive example-based illustration of the functionality of StochasticAD.jl and stochastic triples.
> > 1- In 4.1, what value(s) of $p$ were used? Why not showing in Fig.5 (center) the standard deviation over multiple values of $p$?
>
> As described in the text, we choose $p$ to equal the number of steps so that the transition function has interesting behavior over the range of the walk. In other regimes (e.g. $p$ very small), we observe the same high-variance scaling of the score function.
> > 2- In 4.2 the program is almost deterministic with $p=0.05$. What happens as $p$ approaches 0.5?
>
> (We have revised this response, as our previous reply misinterpreted the reviewer's question.) In our investigation, we have considered $p = 0.5$, i.e. the board starts off at approximately 50% alive and then evolves randomly. The "5%" in the paper refers to the Game of Life rules applied at each step: replacing this with "50%" would mean that there is no difference in the game rules for alive or dead cells, which makes the example less interesting (as the probability approaches 50%, the program essentially randomly flips cells). Note that because of the high number of cells in the game (144), and the fact that the board starts in a completely random state, the full program is highly nondeterministic (the end state of the board is essentially completely different each run.)
>
> > Why not applying the score-function method to the tasks 4.2 and 4.3?
>
> To the best of our knowledge, the score function cannot be used for unbiased AD of the particle filter. At least with the implementation in the recent framework Storchastic, the program must be expressed as a stochastic computation graph, which cannot express the resampling step of a particle filter. Thus we do not know of an AD implementation of the score-function method from which an empirical comparison can be made. For the game of life, based on preliminary experiments, we find that the variance of stochastic triples is substantially lower than a score function implementation (10x lower for the parameters in the paper, thus requiring 100x fewer samples for the same accuracy), and we would be happy to include a more detailed comparison in the final version.
>
> > It is also not clear to me why some approaches mentioned in the Related Work section were not considered (REBAR, RELAX, or the reparametrization trick with Gumbel-softmax).
>
> The Gumbel-Softmax trick cannot be automatically applied to the particle filter or the Game of Life examples, as the discrete structure present in these cases cannot be automatically relaxed (e.g. we cannot do array indexing with a floating point index). We would be happy to implement it for the random walk in the final version. We implemented REBAR and RELAX using the Storchastic framework, but found that they suffer from an exponential scaling issue that forbids their use in AD for our examples.
>
> > 4- In Fig.6 (center), why not reporting both forward and reverse mode AD, possibly using the same colors in Fig.6 (right)?
>
> Forward- and reverse-mode AD of the particle sampler result in an identical distribution of gradient samples. For the system size used in Fig. 6 (center), forward-mode AD is significantly faster than reverse-mode AD. For these reasons, we decided to show only samples from forward-mode AD.
>
> > 5- The paper mentions some additional complications when dealing with multivariate functions. Why not reporting experiments in that sense?
>
> This is a technical detail that needs to be considered when adding, multiplying, etc. stochastic triples, and so this is evaluated in all experiments involving stochastic triples. In our new example-based description, we have simplified the description and we give a concrete example of how a perturbation is probabilistically selected between a Binomial and a Bernoulli.
> > When is a stochastic program "sufficiently regular"?
>
> We provide technical conditions in our formal treatment (Appendix B): the program and its derivative must have bounded expectation and must also satisfy Assumption B.2, which informally says that the possible finite perturbations to a program must be representable by a random variable $Y$. We have updated the main text to refer to Appendix B for these conditions.

---

> > ### Author Response · Authors · 2022-08-02
> > **Response to Reviewer x6td (smaller clarifications)**
> >
> > > "It is convenient to assume probability densities [...] and transition probability densities [...] all depending on the parameters." Why? You mean from a notational perspective?
> >
> > Indeed we had a notational perspective and the similarity to the typical treatment of Kalman filtering applications in mind. However, we realize that the phrasing can be misleading. In our updated version, we instead state: “in general, we allow probability densities .. and transition probability densities to depend arbitrarily on the parameters”.
> >
> > > Fig. 1: I don’t get it. $X’(p+\epsilon)$ with random number sequence z(2) (bottom) in the caption is $X(p+\Delta p)$ in the figure. What is the relation between $\Delta p$ and $\epsilon$?
> >
> > We thank the reviewer for pointing out this inconsistency. We have updated Figure 1. When using finite differences, one samples an independent copy of the program $X(p+\epsilon)$ with (uncoupled) random number sequence $z_2$ (bottom) for some finite choice of $\epsilon$.
> >
> > > "This variance issue can be alleviated if one could run the perturbed program with the same set of random numbers ..". Shouldn't it be 'solved' rather than 'alleviated'?
> > > Def. 2.2 "right (left) derivative" 'stochastic' is missing. Typos: "algorithm. enabling", "an weighted", "with parametrized by", "particle filterunbiasedness"
> >
> > We have fixed these in our revised version.

---

### Official Review · Reviewer_kBKn · 2022-07-12

**Rating:** 6
**Confidence:** 3
**Soundness:** 3 good
**Presentation:** 3 good
**Contribution:** 3 good

**Summary:**

The paper introduces a technique named “stochastic derivative” that helps construct a stochastic program whose expectation is the derivative of the expectation of a given stochastic program, which might include discrete random variables. The technique is then shown to give unbiased estimators and low variance (compared to the baseline score-function method) through three examples: inhomogeneous random walk, stochastic game of life, and particle filter.

**Questions:**

+ It would be helpful if the authors give a list of parameterizations of distributions used in the main paper, like Exp, Bin.
+ Line 87: I wonder what is the definition of Z?
+ Line 105: Why the value of dX(z) is O(eps), I thought this quantity is -eps * log(1-z) which is not bounded when z near 1?
+ Figure 2: The red dashed line indicates a sample obtained by black-box finite differences. What does O(1) imply at this dashed line?
+ Lines 119-120: Why the variance of those independent samples z1, z2 is of order infinity? In addition, it might be more consistent with the introduction section to use X(p + eps) - X(p), rather than X(p + eps/2) − X(p − eps/2).
+ Line 145: Does f need to be bounded? In later examples, I’m seeing f(x) can be x^2 or x^3, which are not bounded.
+ Line 152: typo: infinitely. What does “infinitely close to 1” mean? Why is the importance weight close to 1 here?
+ Line 162: typo: occurring
+ Lines 160-164: I don’t understand this paragraph. Could the author add more details? What does “event of two finite perturbations occurring simultaneously” mean?
+ Line 172: typo: parameterized
+ Line 173: please provide which section in the supplemental that we should look at
+ Figure 4: how can we interpret the output in this figure?


**Limitations:**

Nothing to report.

**Strengths And Weaknesses:**

Overall, I think this is an important paper which introduces a novel technique but needs a major revision for clarity.

The paper is technically sound. I really like that the authors provide detailed motivations and illustrations for each idea and want to thank the authors for that. However, the structure is a bit hard to follow. For example, Section 2 talks about “composable stochastic derivatives” but I only get what “stochastic derivatives” means until Theorem 2.3. Or the section 2.4 talks about Smoothing but lacks background for why it is needed (is this related to Smoothed perturbation analysis mentioned in the introduction?)

I’m not familiar with Julia but would like to reproduce some results in the Demonstrations section in Python to better understand the technique. However, the paper lacks an outline on how to design that stochastic AD system.

The proposed approach is only compared to the score-function method in the inhomogeneous random walk example, showing that using stochastic triples the variance of unbiased gradient estimates. Could the author also provide empirical comparisons to the score-function method in other examples?

**Updated:** I'm happy to increase my score given that the writing quality has been much improved in the last revision.

---

> ### Author Response · Authors · 2022-08-02
> **Response to Reviewer kBKn**
>
> We thank the reviewer for their insightful and thorough comments. We are glad that the reviewer liked our emphasis on motivation and intuition and we appreciate the positive feedback on the figures. Based on the reviewer’s feedback, we have revised the paper to improve its clarity and address their concerns, and also written draft user and developer documentation that is provided in the revised code package submission.
>
> > For example, Section 2 talks about “composable stochastic derivatives” but I only get what “stochastic derivatives” means until Theorem 2.3. Or the section 2.4 talks about Smoothing but lacks background for why it is needed (is this related to Smoothed perturbation analysis mentioned in the introduction?)
>
> We have revised Sections 2 and 3 in this regard. We now explain the term “stochastic derivative” earlier, introduce Theorem 2.3 itself near the beginning of Section 2.2, and provide a self-contained description of smoothing, including the formal definition and why it enables reverse-mode AD. We also hope that our bulleted suggestions in our general response about introducing more mathematically unambiguous statements will help to address your concerns on clarity.
>
> > I’m not familiar with Julia but would like to reproduce some results in the Demonstrations section in Python to better understand the technique. However, the paper lacks an outline on how to design that stochastic AD system.
>
> We have included a walkthrough of a toy usage of `StochasticAD.jl` in the main text to make the essential ideas clear and accessible. We cite and refer to more of the employed implementation concepts from traditional AD systems, such as dual numbers, at the appropriate places. If desired, we would be happy to dedicate an appendix of the paper to the package’s technical implementation in the final version, based on the package’s documentation.
>
> > The proposed approach is only compared to the score-function method in the inhomogeneous random walk example, showing that using stochastic triples the variance of unbiased gradient estimates. Could the author also provide empirical comparisons to the score-function method in other examples?
>
> To the best of our knowledge, the score function cannot be used for unbiased AD of the particle filter. At least with the implementation in Storchastic, the program must be expressed as a stochastic computation graph, which cannot express the resampling step of a particle filter. Thus we do not know of an implementation of the score-function method from which an empirical comparison can be made. For the particle filter, we do compare to previous work that employed the reparameterization trick in a biased manner by neglecting the resampling step; as a generalization of the reparameterization trick, our method naturally extends this previous work by handling the resampling step in an unbiased manner.
> For the game of life, based on preliminary experiments, we find that the variance of stochastic triples is substantially lower than a score function implementation (10x lower for the parameters in the paper, thus requiring 100x fewer samples for the same accuracy), and we would be happy to include a more detailed comparison in the final version.

---

> > ### Author Response · Authors · 2022-08-02
> > **Technical clarifications/revisions from Reviewer kBKn feedback**
> >
> > > It would be helpful if the authors give a list of parameterizations of distributions used in the main paper, like Exp, Bin.
> >
> > We have provided an additional table (Table 1) in Appendix A.1 with the probability distribution functions and cumulative distribution functions (CDFs) of the discrete random variables. The parameterizations of these distributions can be directly expressed as the inverse of these CDFs, and readers may easily verify the stochastic derivatives (Table 2) using our provided formulas.  (We did not supply the explicit form for the Binomial as its stochastic derivative may be expressed automatically via the composition of Bernoullis.)
> > > Line 87: I wonder what is the definition of Z?
> >
> > $Z$ is the sample space for the random variable, from which the random sample $z$ is chosen. We have revised the text to clarify this.
> >
> > > Line 105: Why the value of dX(z) is O(eps), I thought this quantity is -eps * log(1-z) which is not bounded when z near 1?
> >
> > A more precise statement is that $\mathrm{d}X(\varepsilon)$ is point-wise $\mathcal{O}(\varepsilon)$ at each fixed sample $z$ of the sample space Z, although the coefficient may depend on $z$. We have clarified this in the manuscript, and also provided an expression for the event $A(\varepsilon)$ that rigorously defines what it means to have a finite perturbation.
> >
> > > Lines 119-120: Why the variance of those independent samples z1, z2 is of order infinity? In addition, it might be more consistent with the introduction section to use X(p + eps) - X(p), rather than X(p + eps/2) − X(p − eps/2).
> >
> > We have implemented your suggestion, thank you. The samples $X(p+\varepsilon)(z_2)$ and $X(p)(z_1)$ are independent, and hence the variance of their difference approaches $2 \operatorname{Var}(X(p))$ which is non-vanishing even as $\varepsilon$ approaches 0. Since the denominator $\varepsilon$ approaches 0, the variance of the full estimator approaches infinity.
> >
> > > Figure 2: The red dashed line indicates a sample obtained by black-box finite differences. What does O(1) imply at this dashed line?
> >
> > This illustrates the size of the numerator in the finite difference estimate, as described above. We have revised the figure to say $\gg \varepsilon$ instead of $\mathcal{O}(1)$ as this is more informative, and we have also clarified the use of big-$\mathcal{O}$ notation at the beginning of Section 2.
> >
> > > Line 145: Does f need to be bounded? In later examples, I’m seeing f(x) can be x^2 or x^3, which are not bounded.
> >
> > Boundedness is only required for the "test" functions that we use in our definition of the stochastic derivative. These are just a theoretical tool for enforcing a sufficiently strict definition (a similar idea to the concept of "weak convergence" of random variables). For our actual composition result, we have a much weaker requirement, namely that the derivative is bounded in expectation, which permits polynomial functions of random variables such as $\operatorname{Geo}(p)^3$.
> >
> > > Line 152: typo: infinitely. What does “infinitely close to 1” mean? Why is the importance weight close to 1 here?
> >
> > We have revised the text to remove this imprecise language and provided the formal constructions of $w$ and $A$ instead. The mathematical interpretation of this sentence is then that $\mathbb{P}(A(\varepsilon))$, the probability of an infinitesimal change, is differentiable w.r.t. $\varepsilon$ at $\varepsilon = 0$ (so that the complement of this event occurs with probability that approaches 1 as $\varepsilon \to 0$).
> >
> > > Lines 160-164: I don’t understand this paragraph. Could the author add more details? What does “event of two finite perturbations occurring simultaneously” mean?
> >
> > Consider a stochastic program composed of two independent elementary stochastic programs $X_1$ and $X_2$. The probability of a finite perturbation in one program (meaning its output jumps, as characterized by the event $A$) is $\mathcal{O}(\varepsilon)$. So the probability of a jump in both is $\mathcal{O}(\varepsilon)^2$, which we can neglect as $\varepsilon \to 0$ because it contributes a term of size $\mathcal{O}(\varepsilon^2)$ to $\mathbb{E}[\mathrm{d} X(\varepsilon)]$. In our revised Section 2, we are able to explain this more straightforwardly.
> >
> > > Line 173: please provide which section in the supplemental that we should look at
> >
> > This derivation is in Example A.1 of the supplement. We have updated the phrase accordingly.
> >
> > > Figure 4: how can we interpret the output in this figure?
> >
> > The output is printed as an expression "x + δε + (Δ with probability wε)", where x is a sample of the random variable $X(p)$ corresponding to the primary evaluation, and δ, w, and x+Δ are samples of the components $\delta$, $w$, and $Y$, respectively, of the stochastic derivative of $X$ with respect to $p$, according to Definition 2.1. We have made significant efforts to improve the presentation of stochastic triples in Fig. 3 and Section 3.1 of our revised version.

---

### Official Review · Reviewer_bKdb · 2022-07-15

**Rating:** 7
**Confidence:** 2
**Soundness:** 3 good
**Presentation:** 3 good
**Contribution:** 3 good

**Summary:**

This paper proposes a method to compute a derivative of stochastic programs which have discrete dependencies on their parameters. The paper introduces stochastic triples generalizing dual numbers to programs with discrete randomness. The resulting AD method is shown to produce unbiased derivative estimates with lower variance compared to score function methods. A working implementation is developed in Julia. Forward mode and reverse mode AD (using the smoothing technique) is demonstrated on 3 examples. The experiments demonstrate the utility of stochastic triples for unbiased, lower-variance derivates in forward and reverse mode AD on 3 examples.

---

UPDATE: I thank the authors for their detailed responses. After reading the other reviews and author responses, I'm more positively inclined towards the paper and have upgraded my score to "Accept".

**Questions:**

1. Can you provide more detail about how the proposed implementation compares with existing popular reverse-mode AD libraries? What classes of popular applications of reverse-mode AD can StochasticAD.jl be used / not used?





**Limitations:**

Yes

**Strengths And Weaknesses:**

Strengths
  + The paper tackles an important and challenging problem of AD in programs with discrete randomness.
  + The paper makes novel algorithmic contributions (stochastic triples, smoothing) as far as I can tell.
  + The paper includes working code and demonstrates the unbiased and low-variance properties empirically.
  + The paper is intuitively clear and well written.

Weaknesses
  - The novelty of the technical contributions compared to existing AD frameworks could be made a bit clearer. Also, a deeper discussion of which program classes or constructs are not differentiable by popular reverse-mode AD frameworks might help make the contributions clearer.
  - A deeper empirical investigation would have been good to see. In particular, I'd have liked to see experiments demonstrating the limits of the proposed approach as well as the use of "larger" applications instead of the toy examples.

Overall, this seems like a nice paper making progress on a hard problem. I didn't detect any technical issues with the paper but I don't have a deep background in this research area. I'm open to adjusting my scores based on the other reviews.

---

> ### Author Response · Authors · 2022-08-02
> **Response to Reviewer bKdb**
>
> We thank the reviewer very much for their insightful comments. We are glad that the reviewer liked the paper’s ideas and that our efforts to emphasize intuition have indeed been helpful for them.
>
> > The novelty of the technical contributions compared to existing AD frameworks could be made a bit clearer.
>
> We have revised the related work section to emphasize this point. The key novelty of our approach is to generalize the widely-used reparameterization trick to discrete programs while preserving its essential qualities of coupling, unbiasedness, and composability. This is a qualitatively new approach to the problem of stochastic AD, as noted by reviewers kBKn, x6td, and kqcs. In the revised manuscript, we have emphasized the innovations of our approach compared to previous approaches based on continuous relaxations of the program. Specifically, these methods introduce a variance-bias tradeoff and furthermore are not applicable to a number of the discrete constructs we consider in our demonstrations (we discuss this further in our response to your question about this).
>
> Our emphasis on efficient scaling behavior and support for general-purpose composability is another key benefit: the recent Storchastic framework implements an exhaustive set of gradient estimators including score-function based methods such as REBAR and RELAX, but when implementing these methods in Storchastic for the random walk we found that they suffered from exponential scaling issues when composed, making them impractical. In contrast, our demonstrations of forward-mode and reverse-mode AD both have *constant-factor* overhead (around 3 for the random walk, which is highly competitive). Most importantly, this is the same scaling behavior as standard forward-mode and reverse-mode AD.
>
> > Also, a deeper discussion of which program classes or constructs are not differentiable by popular reverse-mode AD frameworks might help make the contributions clearer.
> > Can you provide more detail about how the proposed implementation compares with existing popular reverse-mode AD libraries? What classes of popular applications of reverse-mode AD can StochasticAD.jl be used / not used?
>
> Currently, the only way mainstream reverse-mode AD frameworks can incorporate discrete randomness is through a continuous relaxation of the program. However, if an integer-valued quantity (e.g. the output of a Bernoulli) is relaxed into a floating point value, it cannot be used for e.g. array indexing, whereas in `StochasticAD.jl`, integers stay integers. Thus, a major issue with the continuous relaxation approach is dealing with highly discrete constructs, such as a discrete probability choice based on being alive or dead in the Game of Life or the particle filter resampling step. We have provided an explanation of this at the end of the Game of Life example.
>
> A limitation of `StochasticAD.jl` is that certain control flow constructs like while loops based on inequality conditions cannot yet be automatically differentiated, as we have noted in the updated “limitations and outlook” section. However, existing reverse-mode AD methods cannot automatically handle these constructs either, or indeed even the simpler discrete constructs that `StochasticAD.jl` is able to handle such as array indexing. (The other key limitation is that reverse-mode AD requires smoothing, as you have already noted; deriving a reverse-mode AD algorithm based on unsmoothed stochastic derivatives is an important open problem that will be the subject of future research.)
>
> > A deeper empirical investigation would have been good to see. In particular, I'd have liked to see experiments demonstrating the limits of the proposed approach as well as the use of "larger" applications instead of the toy examples.
>
> In our opinion, the game of life and particle filter example represent very challenging applications for previous techniques due to the discrete structure of the game of life board and particle resampling step. Differentiating such programs would be of great practical utility (see the end of our general response). Furthermore, these examples are high-dimensional and involve a large number of time steps, posing a challenging problem for variance reduction and coupling. We would be happy to provide a more thorough analysis of the tradeoffs and limitations these examples illustrate in the final version.

---

### Author Response · Authors · 2022-08-02
**General Response to the Reviewers**

We thank the reviewers for their thoughtful feedback on our paper. We are happy that the reviewers think that the manuscript presents a novel approach to stochastic automatic differentiation (bKdb, kBKn, x6td, kqcs) and that it addresses an important, impactful, and challenging problem of differentiating discrete randomness (bKdb, kBKn, kqcs).

A common point of feedback was that, while the essential ideas are intuitively clear and well-motivated (bKdb, kBKn), the technical clarity of the paper can be improved. We have made a number of textual modifications, all present in the rebuttal revision and detailed in our specific reviewer responses, which we believe significantly improve the paper’s clarity and address these concerns. Particularly important are the changes we propose to achieve a tighter link between the main text's intuitive treatment and the underlying formalism,
* lifting the key expressions for the weight $w$ and the event $A$ from the appendix into the main text, expressing them in the language of limits as a small change $\varepsilon$ approaches 0,
* providing the full, rigorous form of our composition result in the main text,
* providing the formal definition of the smoothed stochastic derivative in the main text.

With these changes, we can provide a mathematically unambiguous description of our key objects while preserving the intuitive emphasis of the original text.

In a similar spirit, we have worked with the reviewers' feedback on making the AD method as clear and accessible as possible by writing a step-by-step walkthrough of the operation of `StochasticAD.jl` to show how stochastic triples propagate through a multi-step computation, complete with runnable code.

We would also like to emphasize why stochastic derivatives offer a *qualitatively* new contribution to the field. Stochastic derivatives demonstrate how to couple à la the reparameterization trick while exhibiting the two key attributes of *unbiasedness* and *composability*. This has a number of important implications:
* We produce a natural generalization of standard AD: both deterministic AD and the well-known reparameterization trick are *special cases* of our method. Our method composes automatically with well-known AD techniques for handling continuous randomness, and at the same time handles new discrete constructs (e.g. the particle resampling step) where even the score function does not apply.
* We can consider a setting where arbitrary discrete random variables, such as Bernoulli, Binomial, and Geometric variables, may be composed alongside arbitrary continuous randomness, at no loss of speed. Further, we demonstrate our approach on examples with fundamentally discrete structure (e.g. the Game of Life), where popular continuous relaxation approaches are no longer applicable.

In light of the above, we feel that the developed methodology is not solely an empirical improvement, but opens the door to automatic stochastic gradient estimation in fundamentally new settings with greater discrete structure. We note that these examples have considerable practical utility. For example, many scientists are looking to mix machine learning (ML) with agent-based models found in epidemiology and sociological contexts, and these models generally use programming constructs similar to our stochastic Game of Life. Systems biology uses many rule-based models and discrete stochastic (Gillespie) simulations of discrete chemicals which have similar limitations, restricting their use as part of ML pipelines. Our methodology extends differentiable programming to these crucial applications in an unbiased and low-variance way.

---

### Meta-Review · Area_Chair_Qnsd · 2022-08-26

**Recommendation:** Accept
**Confidence:** Less certain

**Metareview:**

The topic of this paper is interesting, and I wanted to get the basic idea, so I looked at the paper myself.  Had I been a reviewer I would have recommended rejection.  However, as the existing reviews are positive, and the authors will not have a chance to respond to my objections, I will recommend acceptance (in spite of the low confidence scores of the reviewers where the high scores have confidence 2.) For the sake of the authors I will list my complaints.

My fundamental complaint is about clarity.  The central problem is the notation X(p).  In the main text this is only defined as a "stochastic program".  But what is it allowed to be a distribution on?  What is the allowed dimension of the parameter (vector?) p? If X(p) is an arbitrary stochastic program what is E[X(p)]?  If X(p) defines a distribution on a finite abstract set then clearly E[X(p)] CANNOT mean the expectation of a random value.  The first sentence of appendix B.2 need to appear before any appearance of the notation E[X(p)] in the main text. Even if that is done, the authors are assuming a fixed embedding of each random value of x in a Euclidean space.  But in the vast majority of modern applications that embedding must be learned.  For example, in a VAE model of grammar induction one must learn an embedding of the grammar nonterminal symbols. The formal set up would be greatly clarified by explicitly introducing a "loss function" f as part of the given data and write, for example E_{x \sim X(p)} [f(x)].  The main body of the text needs more intuition about the meaning of w and Y in theorem 2.2 and some kind of sketch of a proof.  The paper would be significantly simpler if limited it to the case where X(p) is entirely discrete.  This is the interesting case and there is then no need for X'(p). The technically hard issue is w and Y in theorem 2.2.  If a mixed stochastic-continuous case is needed for the general case then it could come later where it is well motivated.    An explicit form for w and Y should appear in the body of the paper rather than a naked claim that they exist.  Any implementation must construct w and Y so the proof needs to be constructive.  Examples are nice, but they are no substitute for the proof of the general case.  Some sketch of that proof needs to appear in the body of the text --- most importantly a computable solution for w and Y. As it stands it appears that the authors are trying to hide the artificial nature of their technical set-up --- the use of a-priori fixed embeddings of abstract tokens.



**Award:**

No

---

### Decision · Program_Chairs · 2022-09-14

Accept